# A planetary health perspective on the translation of climate change research into public health policy and practice: A scoping review protocol

**Chanelle Mulopo**[1]*, **Samuel Abimbola**[2], **Nyamongo Onkoba**[3], **Bey-Marrie Schmidt**[1,4]

1 School of Public Health, University of the Western Cape, Cape Town, South Africa, 2 Cyprus International Institute for Environmental and Public Health, Cyprus University of Technology, Limassol, Cyprus, 3 School of Health and Biomedical Sciences, The Technical University of Kenya, Nairobi, Kenya, 4 Health Systems Research Unit, South African Medical Research Council, Cape Town, South Africa

* cmulopo@uwc.ac.za

**Funding:** The work reported herein was made possible through funding by the South African Medical Research Council through its Division of Research Capacity Development under the SAMRC

## Abstract

### Background

Climate Change (CC) emanating from anthropocentric human activities is a great threat to the quality of human life and well-being worldwide. The translation of CC research evidence can play a critical role in promoting the formulation of climate-sensitive policies to equip public health systems for CC-associated disaster preparedness, response, and management. This scoping review seeks to explore knowledge translation approaches for promoting, the uptake, and use of CC research evidence in public health policy and practice.

### Methods

This scoping review will be conducted according to the guidelines of Arksey and O'Malley. A search strategy will be developed for published articles in PubMed, CINAHL, and Scopus databases and for grey literature in the World Health Organization, Planetary Health Alliance, and the University of the Western Cape repositories.

### Discussion

The proposed scoping review will gather existing evidence on the relationship between knowledge translation, CC research, and public health decision-making. This will provide insights into research and practice gaps, and recommendations will be made to ensure effective knowledge translation for CC related decision-making.

## Background

Anthropogenic activities have an impact on planetary health–the health of people and the state of the natural environment where they live and on which they depend [1]. Planetary health recognizes the health of the planet as a system in which human health and the health of the

Research Capacity Development Initiative-
Postdoctoral Fellowship Programme from funding
received from the South African National Treasury.
The contents hereof are the sole responsibility of
the authors and do not necessarily represent the
official views of the SAMRC or the funders.

**Competing interests:** The authors declare that they
have no competing interests.

planet are inextricably interconnected [2]. Planetary health includes climate change (CC), ocean acidification, land degradation, water scarcity, overexploitation of fisheries, and biodiversity loss [3, 4]. Globally, CC, land degradation, and biodiversity loss are argued to have the most devastating impact on animal and human health [3, 5–7]. For instance, many human and animal deaths have been attributed to CC as a result of increases in temperatures, floods, droughts, exacerbation of chronic diseases and maternal health outcomes, and changing patterns of infectious diseases among vulnerable communities due to the shortened reproductive cycle of vectors and pathogens, vector migration and change in habitat of vectors in response to CC, among others [6, 8, 9]. Although not all heat-related deaths (356000 worldwide) are attributed to CC [5], Vicedo-Cabrera *et al.* [7] reported that more than one-third of all heat-related deaths recorded during 1991–2018 were linked to CC. Moreover, by the end of the century, it is estimated that CC will cause 3.4 million deaths per year [10]. If the statistics of Vicedo-Cabrera *et al.* [7] hold true for the remainder of the century, then an estimated 1.1 million deaths will be attributable to climate-related heat stress by the end of the century.

Several studies have been conducted to establish the effects of CC in terms of climate risk management and food security, human dimensions and health, policy and communications, technologies, water environments, and ecosystems [11]. However, CC research knowledge has not been translated into tangible solutions for public health policies and practices. Therefore, there is an urgent need for evidence-based solutions on CC research for preparedness, response, and management of CC-associated disasters in low- and middle-income countries (LMICs). Thus, the integrated knowledge translation (IKT) approach is a strategy that can be utilized to facilitate the formulation of public health policies and guidelines for decision-making in CC and health. IKT is an approach that facilitates collaboration (in the form of knowledge exchange, dialogue, and capacity building) between relevant decision-makers and researchers [12]. Knowledge exchange refers to the co-production and dynamic exchange of information between relevant stakeholders designed to result in mutual benefits for all parties [13]. Dialogues involve initiating discussions with a diverse group of policymakers, stakeholders, and researchers, where sometimes an evidence brief and dialogue summary of the best available research evidence is circulated beforehand to decision-makers to stimulate thoughts and discussions [14]. Capacity building refers to initiatives dedicated to improving (a) the researcher's knowledge of decision-making processes [15] and (b) policymakers' and stakeholders' skills to find and use research evidence on their own [14]. The importance of knowledge translation to the overall improvement of policy and practice in any given field of human endeavour is continuously being evidenced by research findings. Grimshaw *et al.* [16] concluded that while there is immense benefit in translating research findings into policy and practices in the health sector, the evidence of such application of knowledge is presently very weak and warrants further research. For example, Lavis *et al.* studied eight policy-making processes in Canada and reported that only one had citable research used in all stages of the process [17]. Furthermore, there has been inconsistent use of evidence by World Health Organization (WHO) policymakers [18]. Knowledge translation research involves exploring and measuring gaps in decision-making, improving knowledge synthesis and distillation, enhancing the diagnosis and measurement of determinants of knowledge uptake, and determining the effectiveness and sustainability of different knowledge translation approaches [19]. The IKT approach could be essential in identifying and bringing all relevant CC stakeholders together to engage in knowledge exchange, dialogue, and capacity building that can reduce the impact of CC and its associated risks on the population and the health system. Additionally, the IKT approach could contribute to improving public health surveillance systems, sensitization to detect potential effects of climate change, improving infectious disease surveillance systems, and enhancing awareness of CC among public health and medical practitioners [6].

There is a need to understand whether, how, and where knowledge translation approaches are implemented to promote the uptake and use of CC research for health. In recent years, many studies have linked CC and health [20, 21]; however, we have not identified any review on decision-making in relation to CC and health. Hence, there is an opportunity to synthesize literature on this topic to inform both decision-makers and researchers working in the field of CC and health on how the best evidence on CC can be promoted in decision-making.

The importance of knowledge translation in public health cannot be overemphasized. It is pertinent that research be put to practical use in addressing real-world problems and enhancing the overall health outcomes of populations. It also ensures that research is accessible and understandable to the public, policymakers, and practitioners, helping them make informed decisions and encouraging evidence-based practices. In summary, knowledge translation is crucial in science to ensure that research findings are not confined to academic journals, but are applied, understood, and used for the betterment of society. It facilitates evidence-based decision-making, innovation, and the advancement of fields such as public health, medicine, and environmental science. To the best of our knowledge, there is a paucity of data on the knowledge transition of CC research, not only in South Africa but also in Africa as a whole. The authors hope to address this gap by synergizing existing data in a way that research and policy gaps can become easily identifiable and utilized. As the authors' backgrounds are diverse, this scoping review will embrace multiple schools of thought, setting the foundation for cross-disciplinary collaboration in addressing CC challenges in the global south.

## Identifying the research questions

This scoping review aims to identify and synthesize evidence on the relationship between knowledge translation, CC research, and health policy and practices, and decision-making from a planetary health perspective. The specific objectives of the scoping review are as follows:

- To explore the relationship between knowledge translation, CC research, and health decision-making.

- To identify whether, how, and what type of CC research is being translated into health decision-making.

- To explore the knowledge translation approaches being used or implemented to promote the uptake and use of CC research in health decision-making.

The population context content framework will be applied to this study; the focus of the population is global, the context is knowledge translation, and the content is CC and planetary health.

## Identifying relevant studies

This scoping review will be conducted according to the methods proposed by Arksey and O'Malley [22]. Conducting the scoping review will involve identifying the research question, identifying relevant studies, selecting studies, charting the data, collating, summarizing, and reporting the results, with consultation as an additional element. The strength of Arksey and O'Malley's scoping review methodology is that it provides a systematic and transparent approach for mapping evidence in a particular area of interest. However, the limitation of this methodology is that it does not provide guidance on the iteration between steps and how to manage a large number of search records [22]. The scoping review findings will be reported

using the PRISMA Extension for Scoping Reviews (PRISMA-ScR): Checklist and Explanation [23] as described in the **S1 File**. We aim to conduct the review for a period of 12 months.

The scoping review will include literature on knowledge translation, CC research, and health decision-making. Peer-reviewed research studies (with no methodological restrictions), and grey literature, are eligible. The search will identify all relevant studies from 2003 to 2023. The focus of this review is studies conducted over the last two decades. The main reason is that there has been much attention and focus on the relationship between CC and health in the last two decades [24–26].

The main author (CM), will conduct the litterature search in the following electronic databases for eligible studies: PubMed, CINAHL, and Scopus, and repositories such as the WHO, the Planetary Health Alliance, and the University of the Western Cape. Search terms in PubMed will include the distillation of keywords and Medical Subject Headings (MeSH) terms related to knowledge translation (concept A) (e.g., knowledge uptake, knowledge utilization, knowledge use, and knowledge transfer), CC (concept B) (e.g., climate change, global warming, and climate variability) and health. The search will identify all relevant studies without geographic restrictions but will include restrictions on the date and language. Only studies conducted in English from 2003 to 2023 will be included in the scoping review. We have developed a preliminary search strategy using relevant keywords and MeSH terms (**S2 File**). To ensure that potential studies are not missed, we will apply an iterative approach using known studies that meet the inclusion criteria identified during the preparation of the protocol. Three examples of studies meet the inclusion criteria for the current scoping review: Tchoukaleyska *et al.*, 2021 [27] Fears *et al.*, 2021 [28], and Lapaige and Essiembre, 2010 [29]. Studies that meet the inclusion criteria will be searched for among "hits" (search records) and used to identify new keywords and MeSH terms not already included in the search strategy. Once the search strategy has been finalized using the PubMed database, we will tailor it to each database and report on the adaptations.

Four reviewers (CM, SA, NO, and BS) will independently screen the titles and abstracts to evaluate their eligibility for full-text review. The full texts of those studies identified as potentially relevant will be retrieved and read by the two reviewers to make a final decision about inclusion. During this full-text review stage, where necessary, study authors will be contacted for further information. At both the abstract and full-text screening stages, conflicts will be resolved by the two reviewers (CM and NO) first attempting to reach a consensus view; failing which, a third reviewer (BS) will be the final arbitrator.

In addition to the electronic searches, the review authors (CM and SA) will (a) search the reference lists of all included studies and key references (e.g., relevant systematic reviews) and (b) contact authors of included studies and/or experts in the field for additional references.

## Study selection

An initial search from different databases will be conducted to identify relevant studies using abstracts and titles. The search results will be collated in the Endnote Reference Manager (Version 20.6), and duplicates removed [30]. The final search database will then be uploaded into Covidence software (https://www.covidence.org/), a web-based collaboration software for screening titles, abstracts, data extraction, and monitoring the collaborative process in the production of systematic reviews [31].

The following inclusion criteria will be used:

(a) Types of studies:

All studies published from 2003 to 2023 involving stakeholders working on knowledge translation and CC research will be included in the review. We are interested in studies that

describe the relationship between different stakeholders relevant to knowledge translation and climate research policy and practices. Relevant stakeholders include participants and researchers in the CC field; these could be researchers affiliated with universities or research centers or any climate/environmental network or government. Other participants to be included will be practitioners such as medical doctors, public health specialists, practitioners in the environment and health sector, and policymakers responsible for developing policies on CC and health decision-making. Additionally, CC activists in NGOs, Community-Based Organizations (CBO), and any other civil society movements that are interested in CC and health are of interest to us. Lastly, "others," referring to stakeholders in the private and public sector involved or affected by CC research in decision-making.

(b) Setting

The scoping review will take on a global perspective to explore the translation of CC research into policy and practices. The outcome will indicate how CC research can be used to inform policy and practice in health decision-making. Furthermore, the review will highlight the extent of knowledge translation in CC research and how much of the evidence is being used to inform health decision-making. The underlying regional challenges underpinning the lack of research-driven and evidence-based policies and decision-making within this space will also be highlighted.

(c) Charting the data

A data extraction template (S3 File) will be developed and piloted to facilitate the extraction of important information from the studies that will be included in the review. Data extraction will be conducted by the two review authors (CM and SA), who will collect, sift, and sort the data according to the review objectives. Data extraction will be done in Excel to allow for comparison of key items across studies and to allow for synthesis within and across data items. Once all the data has been extracted and checked, studies will be categorized or "charted: according to the following criteria: (a) What is the relationship between knowledge translation, CC research and health decision-making? (b) What type of CC research is being translated in health decision-making? and (c) What knowledge translation approaches are being used or implemented to promote the uptake and use of CC research in health decision-making? The methodological quality of the included studies will not be assessed, as per scoping review recommendations [22].

## Collating summarizing, and reporting the results

The authors will apply the six steps of thematic analysis recommended by Braun and Clarke to analyze the data [32]. Data will be coded and analyzed by one review author (CM) in ATLAS.ti (www.atlasti.com). The project bundle on ATLAS.ti will be shared with a second review author (SA), who will check the coding and data analysis on an ongoing basis. The data from various studies relevant to the research question will be synthesized according to variation (breadth) and key components (depth) pertaining to the types of existing evidence on knowledge translation and CC research in health decision-making; the geographical locations where most studies were conducted as well as the strategies used to translate evidence into policy and action. In the analysis, a combination of quantitative and qualitative syntheses will be conducted to provide an overview of the findings. Quantitative data will be summarized using descriptive statistics. The authors anticipate using qualitative thematic analysis to summarize narratives on knowledge translation in planetary health. Regular meetings and consensus discussions will be held with the aim of eliminating biases and striving to achieve a mutual interpretation of the review findings.

### Consultations

Once the preliminary analysis is complete, we will consult with researchers in the field of CC research to gather additional data on knowledge translation in CC and health decision-making; this will help make sense of the findings. Additionally, we will also contact authors of studies included in the scoping review for additional relevant studies.

## Patient and public participation

There was no patient or public involvement in the design of this protocol.

## Discussion

This scoping review will draw on the planetary health framework where appropriate. The proposed scoping review will gather existing evidence on knowledge translation, CC research, and health decision-making. This will highlight the gaps, and recommendations will be made to ensure effective knowledge translation in relation to CC health decision-making. To our knowledge, this will be the first review that will synthesize studies on knowledge translation strategies, CC, and public health globally. Befitting the fact that planetary health is a new field [33] and considering the surge in the literature on CC in the last two decades [34–36], the need arises to determine whether the research being conducted is translated into policy and practice for it to achieve the necessary impact.

While we investigate knowledge translation strategies in relation to CC and public health globally, the review will paint a clear picture of the patterns of knowledge translation in relation to where the translation is taking place and who is doing the translation. The review article will highlight the geographical disparities that exist in knowledge translation among different global regions.

The review article will also highlight knowledge translation strategies that are seemingly effective and could be applied in regions that are or could be facing challenges with health system adaptation due to the lack of knowledge translation. We foresee a publication bias on this topic towards the global north. Therefore, unpacking the content of what is being translated will be explored in relation to who benefits and how other regions that are more vulnerable to CC are impacted.

## Supporting information

**S1 File. PRISMA-P checklist.**
(DOCX)

**S2 File. Preliminary search strategy.**
(DOCX)

**S3 File. Data extraction template.**
(XLSX)

## Acknowledgments

We would like to thank Mr. Gerald Louw, the subject librarian at the University of the Western Cape, for assisting in developing the search strategy.

## Author Contributions

**Conceptualization:** Chanelle Mulopo, Samuel Abimbola, Bey-Marrie Schmidt.

**Methodology:** Chanelle Mulopo, Bey-Marrie Schmidt.

**Writing – original draft:** Chanelle Mulopo.

**Writing – review & editing:** Chanelle Mulopo, Samuel Abimbola, Nyamongo Onkoba, Bey-Marrie Schmidt.

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
