## [Decision Letter · Decision Letter 0]

26 Jun 2023

PONE-D-23-09445A planetary health perspective on translation of climate change research into public health policy and practice: A scoping review protocolPLOS ONE

Dear Dr. Mulopo,

Thank you for submitting your manuscript to PLOS ONE. After careful consideration, we feel that it has merit but does not fully meet PLOS ONE’s publication criteria as it currently stands. Therefore, we invite you to submit a revised version of the manuscript that addresses the points raised during the review process.

We look forward to receiving your revised manuscript.

Kind regards,

Udoka Okpalauwaekwe, MD, MPH, PhD

Academic Editor

PLOS ONE

Journal Requirements:

Reviewers' comments:

Reviewer's Responses to Questions

**Comments to the Author**

1. Does the manuscript provide a valid rationale for the proposed study, with clearly identified and justified research questions?

Reviewer #1: Yes

Reviewer #2: Yes

Reviewer #3: Yes

2. Is the protocol technically sound and planned in a manner that will lead to a meaningful outcome and allow testing the stated hypotheses?

Reviewer #1: Yes

Reviewer #2: Yes

Reviewer #3: No

3. Is the methodology feasible and described in sufficient detail to allow the work to be replicable?

Reviewer #1: Yes

Reviewer #2: Yes

Reviewer #3: No

4. Have the authors described where all data underlying the findings will be made available when the study is complete?

Reviewer #1: Yes

Reviewer #2: Yes

Reviewer #3: Yes

5. Is the manuscript presented in an intelligible fashion and written in standard English?

Reviewer #1: Yes

Reviewer #2: Yes

Reviewer #3: No

6. Review Comments to the Author

You may also provide optional suggestions and comments to authors that they might find helpful in planning their study.

Reviewer #1: Paper is well written except some avoidable grammatical errors which could have been avoided if authors read through the manuscript well.

Overall, it will be a major contribution to knowledge

Reviewer #2: The protocol is well written. However, some edits and tightening up the search section and the methodology. is needed.

-The significance, rationale and objectives are justified.

-The background – page 4 line 15 ‘Many deaths have been attributed to CC are as a result of increase in temperatures, and changing patterns of infectious diseases among vulnerable communities (6). If necessary explain the link between CC and changing patterns of infectious disease among vulnerable groups. Are there other CC related factors responsible for deaths?

-Include some literature on research knowledge translation.

-Need some editing.

-It is commendable that authors will be guided by Arksey and O’Malley methodology. However, this methodology though offers a good foundation, authors should highlight strengths and potential limitations that are inherent in this methodology.

- Identifying relevant studies: Review time is unclear: 2000 onwards (2000-2022) or last two decades. Please clarify.

- The preliminary search strategy using relevant keywords and MeSH terms (Supplementary File 2), I suggest to add AND NOT to Boolean Operators OR and AND.

-Clarify who is conducting the search- the authors, author reviewers, or reviewers. The study will benefit from a librarian?

-Covidence: Please clarify how Covidence will be used in managing this scoping review?

Reviewer #3: The method proposed to present the study does not contain sufficient detail. Moreover, the language used in the protocol is far below standard English; contains grammatical and typographical errors.

7. PLOS authors have the option to publish the peer review history of their article (what does this mean?). If published, this will include your full peer review and any attached files.

Reviewer #1: **Yes: **Samuel Egykwa Ankomah

Reviewer #2: **Yes: **Emmy Kageha Igonya

Reviewer #3: No

---

## [Author Response · Author response to Decision Letter 0]

28 Nov 2023

Rebuttal Table

Manuscript title: A planetary health perspective on translation of climate change research into public health policy and practice: A scoping review protocol

Reviewer 1 

Comments Response to Comments Changes made in the document

Pg. 4: “In 2019 it was estimated that 356, 000 deaths worldwide were link to extreme heat (7),…..”

(i) Change ‘link’ to linked The authors have changed the word link to “linked” Page 6 line 112

“In 2019 it was estimated that 356, 000 deaths worldwide were link to extreme heat (7),…..”

(i) Not all extreme heat could be attributed to CC. The authors need to situate this statement appropriately in the context of CC

 The authors have substantiated the sentence and situated in the context of climate change. Page 4 line 68-74

“In literature, several have been conducted to establish effects on CC………”

(i) Several studies?? The authors have included the word “studies” Page 4 line 75

“However, CC research knowledge has not been not translated into tangible solution in relation to public health policy and practice”.

(i) Kindly delete ‘not’ before translated 

 The authors have removed the second “not” 

“Integrated knowledge translation (IKT) approach is one strategy that can be utilised to applied”….

(i) To apply?? The authors have removed the word “applied” 

There are places climate change is shortened as “CC” and in some places written fully. I think this should be consistent throughout the paper

 The authors have made the changes in the document and made it consistent by using “CC” throughout the document. Changes made throughout the document.

Pg.5 

‘we have not identified any reviews on decision making in climate change……

(i) We have not identified any review……please delete the ‘s’

 The authors have deleted the “s” 

Page 8

“Any studies…….”

(i) Rephrase to Any study We have rephrased to any review Page 6. Line 112-113

Type of participants

This section does not come clear to readers. This is a scoping review so mentioning study participants such as researchers working in the field of climate change etc does come to clear. Maybe authors need to explain this section and the role of those participants in the scoping review. We have revised this section and provided clarity on what we mean by participants by saying that we are interested in identifying the participants or stakeholders mentioned in the articles Pg. 9-10 Line 202-212

Are the authors using the PCC framework in this regard? More explanation needed or that whole section could be deleted We have included the PCC framework will be applied to the scoping review. Pg. 7 Line 141-143

Can the authors expand the database to Scopus etc. It appears what has been listed is too limiting. We have decided to expand the database to include Scopus Page 8 line 169

Again, will the study include only English Language papers? That needs to be stated clearly We had indicated previously that only studies conducted in English will be included in the study. Page 8 line 170-171 

Will the entire scoping review be guided by a theoretical framework on CC and Health? If so, this must be stated in the manuscript. We will draw on the planetary health framework where appropriate to guide data extraction, analysis and synthesis. 

Overall, I think the authors need to read through the manuscript again and correct some basic grammatical errors which could have been avoided. The authors have had the manuscript proof read. 

Reviewer2 

The significance, rationale and objectives are justified. Thank you, we appreciate the positive feedback. 

-The background – page 4 line 15 ‘Many deaths have been attributed to CC are as a result of increase in temperatures, and changing patterns of infectious diseases among vulnerable communities (6). If necessary explain the link between CC and changing patterns of infectious disease among vulnerable groups. 

Are there other CC related factors responsible for deaths? Thank you for the comment. While it may be an interesting prospect to discuss the link between CC and how it affects infectious diseases mechanism, it is beyond the scope of this review and outside our defined objectives. However, we have added a sentence in line with your suggestion.

A few more factors have been included. Thank you Pg. 4 line 63-68

Include some literature on research knowledge translation. We have included literature on knowledge translation research. Pg. 5 Line 92-104

Need some editing. Thank you for the comment the article has been edited. 

-It is commendable that authors will be guided by Arksey and O’Malley methodology. However, this methodology though offers a good foundation, authors should highlight strengths and potential limitations that are inherent in this methodology. Thank you, we have included the strengths and limitations of conducting a scoping review highlighted by Arksey and O’Malley. Pg.7 Line 149-153

Identifying relevant studies: Review time is unclear: 2000 onwards (2000-2022) or last two decades. Please clarify. We have rectified and stated that we will search for studies from 2003-2023 Pg. 8 Line 159

The preliminary search strategy using relevant keywords and MeSH terms (Supplementary File 2), I suggest to add AND NOT to Boolean Operators OR and AND. WE have revised the search string and added AND NOT to the Boolean Operators OR and AND. The search output has been updated (see Supplementary File 2)

Clarify who is conducting the search- the authors, author reviewers, or reviewers. The study will benefit from a librarian? I have clarified that I (CM)the main author will conduct the initial searches with the assistance of a librarian 

Additionally, the search of the reference list of included studies and contacting authors will be done be myself (CM) and the second author (SA) Pg. 8 Line 162

Pg. 9 Line 187

Covidence: Please clarify how Covidence will be used in managing this scoping review? We have clarified in the manuscript that Covidence will be used to screen, extract data and keep track of the screening. Pg. 9 Line 194-197

Collating summarizing, and reporting the results: explain more on qualitative analysis process. We have stated that we will apply the six steps of thematic analysis by Braun and Clark. Pg. 11 Line 237-240

Reviewer 3 

Pre-testing is necessary for scoping reviews to ensure whether there is existing literature on the research topic. However, the authors failed to produce evidence that their study is feasible.

 Thank you for your comment. We have provided proof that our study is feasible. Kindly see supplementary File 2 that shows that the search string in PubMed yielded 1879 articles

Major issue: 1. The authors failed to explain how their topic is important in the background.

 The authors have added more text to justify why the study is important and why it should be conducted. Pg. 6 line 116-129

Major issue: 2. the methodology lacks some clarity; how data will be reported eg. Authors can propose and adopt the PRISMA flow diagram to report data. Thank you for the comment it was mentioned previously that the PRISMA flow diagram will be used to report the data. Pg. 8 Line 154

Minor issues: 1. authors failed to prof-read the manuscript and that the manuscript contains unacceptable typographical errors and grammatical mistakes e.g. Please check the lines below from the background of the manuscript: Thank you for the comment. The authors have proof-read the entire manuscript. 

63. comprises of – comprise does not take a preposition Thank you, we have removed “of” 

67. CC are as a result of increase – subject and verb agreement (as a result of increases) Thank you we have edited this from increase to increases. Pg. 4 line 64

69. were link – subject and verb agreement (were linked) Thank you, we have worked on this and changed the term from link to linked. Pg. 4 line 70

71. effects on - wrong use of preposition. It should be the effects ‘of’ We have edited accordingly Pg. 4 line 75

73. double negative – not and not We have edited this as per your request. 

78. utilised to applied – to should be followed by a verb, so to apply We have edited this has per your suggestion Pg. 5 line 81-82.

83. benefit of all parties - incorrect preposition (benefit for all parties) We have edited this as per your suggestion Pg. 5 line 86

84. Dialogues involves - wrong use of verb (it should be ‘involve’) We have edited this as per your request Pg. 5 line 87

These mistakes are not exhaustive. All other sections are full of errors and mistakes. We took note of this and proofread the article. 

The authors proofread the manuscript and correct all the avoidable errors. We have done this thank you 

Pretest their questions and if necessary amend the topic per the existing literature We have pre-tested and conducted the search for study feasibility. Kindly see supplementary File 2. Thank you 

Read on the PRISMA flow diagram and capture it in this protocol We have done this. Thank you.

---

## [Editor Report · Decision Letter 1]

4 Dec 2023

A planetary health perspective on translation of climate change research into public health policy and practice: A scoping review protocol

PONE-D-23-09445R1

Dear Dr. Mulopo,

We’re pleased to inform you that your manuscript has been judged scientifically suitable for publication and will be formally accepted for publication once it meets all outstanding technical requirements.

Kind regards,

Udoka Okpalauwaekwe, MD, MPH, PhD

Academic Editor

PLOS ONE

---

## [Editor Report · Acceptance letter]

1 Mar 2024

PONE-D-23-09445R1 

PLOS ONE

Dear Dr. Mulopo, 

I'm pleased to inform you that your manuscript has been deemed suitable for publication in PLOS ONE. Congratulations! Your manuscript is now being handed over to our production team.

Kind regards, 

on behalf of

Dr. Udoka Okpalauwaekwe 

Academic Editor

PLOS ONE